# Integrating Content-Semantics-World Knowledge to Detect Stress from Videos

## ABSTRACT

Stress has rapidly emerged as a significant public health concern in the contemporary society, necessitating prompt identification and effective intervention strategies. Video-based stress detection offers a non-invasive, low-cost, and mass-reaching approach for identifying stress. In this paper, we propose a three-level content-semantic-world knowledge framework, addressing three particular issues for video-based stress detection. (1) How to abstract and encode video semantics with frame contents into visual representation? (2) How to leverage general-purpose LMMs to augment task-specific visual representation? (3) To what extent could general-purpose LMMs contribute to video-based stress detection? We design a Slow-Emotion-Fast-Action scheme to encode fast temporal changes of body actions revealed from video frames, as well as subtle details of emotions per video segment, into visual representation. We augment task-specific visual representation with linguistic facial expression descriptions by prompting general-purpose Large Multimodal Models (LMMs). A knowledge retriever is designed to evaluate and select the most proper deliverable of LMMs. Experimental results on two video-based stress detection datasets show that 1) our proposed three-level framework can achieve 90.89% F1-score in UVSD dataset and 80.79% F1-score, outperforming state-of-the-art; 2) leveraging LMMs helps to improve the F1-score by 2.25% in UVSD and 3.55% in RSL, compared to using the traditional Facial Action Coding System; 3) purely relying on general-purpose LMMs is insufficient with 88.73% F1-score in UVSD dataset and 77.48% F1-score in RSL dataset, demonstrating the necessity to combine task-specific dedicated solutions with world knowledge given by LMMs.

## CCS CONCEPTS

• **Information systems** → *Multimedia information systems*; • **Applied computing** → *Psychology*.

## KEYWORDS

Stress detection, Video, Large multimodal models

## 1 INTRODUCTION

In today's rapidly advancing society, people are experiencing unprecedented levels of stress, stemming from both traditional factors (such as further education, examination, marriage, etc.), as well as newly emerging stressors (such as the outbreak of the coronavirus

*ACM MM, 2024, Melbourne, Australia*
© 2024 Copyright held by the owner/author(s). Publication rights licensed to ACM.
ACM ISBN 978-x-xxxx-xxxx-x/YY/MM
https://doi.org/10.1145/nnnnnnn.nnnnnnn

epidemic and the displacement of jobs by artificial intelligence). Too much stress, if left unchecked, can have profound physical and psychological ramifications, exacerbating existing issues and ensnaring life in a harmful cycle [3, 15]. Recognizing the critical importance of early intervention, detection of stress emerges as a paramount concern [41].

Traditional ways rely on well-defined psychological questionnaires (e.g., Cohen's Perceived Stress Scale (PSS-14) [4] and Social Readjustment Rating Scale (SRRS) [17]), various physiological signals (e.g., blood pressure, heart rate, electromyography, electroencephalogram, etc. [11, 32]), and/or social media behaviors [8, 25, 40–42] for stress detection. Recently, with the wide deployment of surveillance videos, as well as the rapid development of machine learning techniques, detecting stress from videos based on hand-crafted and/or deeply learned features has attracted research attention due to its non-invasiveness, low-cost, and mass-reaching advantages [13, 16, 19, 20, 22, 47, 48].

In this study, we are interested in examining the role of world knowledge delivered by Large Multimodal Models (LMMs) which made significant progress very recently in video-based stress detection. To this end, we build **a three-level (content-semantic-world knowledge) framework**, enclosing three particular questions to be addressed towards stress detection.

$Q_1$: How to abstract and encode video semantics with frame contents into visual representation?

$Q_2$: How to leverage general-purpose LMMs to aid task-specific visual representation, and strengthen video-based stress detection?

$Q_3$: To what extent could general-purpose LMMs contribute to video-based stress detection?

According to the psychological studies, stress is a feeling of emotional strain and pressure [30]. Fostering comprehensive understanding of stress manifestations (emotion and emotion dynamics) could inevitably enhance the stress detection capability due to the high correlation of stressful states and emotions. Hereby, to address $Q_1$, we attempt to sense emotion (of seven possible categories - joy, love, surprised, angry, sorrow, anxiety, and hate) from each video frame to grasp the most prominent emotion per video segment. Specifically, as human emotions exhibit a slower pace of shift than their body actions [24, 31], we draw inspirations from the powerful SlowFast mechanism [12], and design a Slow-Emotion-Fast-Action scheme to encode segment-wise emotion with fast temporal changes of body actions as reflected from each frame in the video segment into visual representation.

Furthermore, as emotions evoke specific patterns of facial muscle actions, but what is expressed by the muscle movements cannot be fully encapsulated by the emotions [2], we augment the obtained visual representation with linguistic facial expression descriptions delivered by general-purpose pre-trained Large Multimodal Model (LMMs). Considering that LMMs tend to hallucinate, and may generate contents unfaithful to the specific requirement [49], we develop

a knowledge retriever to evaluate multiple generations of facial expression descriptions from LMMs based on original face images, and then select the best generation as the final facial expression description to address $Q_2$.

In response to $Q_3$, we evaluate the performance of the proposed three-level stress detection framework on two stress detection datasets. The experimental results show that 1) our method can achieve strong performance with 90.89% F1-score in UVSD dataset and 80.79% F1-score in RSL dataset; 2) leveraging LMMs enables to improve the detection by 2.25% F1-score in UVSD and 3.55% F1-score in RSL, compared to using the traditional Facial Action Coding System (FACS) [10] which decomposes facial expressions into actions appearing on different facial parts, named Action Units (AUs) [23]. 3) purely relying on general-purpose LMMs is insufficient with in 88.73% F1-score in 77.48% UVSD and F1-score in RSL, demonstrating the necessity to combine task-specific dedicate solutions with world knowledge given by LMMs.

The contributions of the paper can be summarized as follows.

- We present a three-level framework, unifying frame-wise video content, video segment-wise emotions, and world knowledge about facial expressions for video-based stress detection.
- We design a Slow-Emotion-Fast-Action scheme to encode fast temporal changes of body actions revealed from video frames, as well as subtle details of emotions per video segment, into visual representation.
- We explore the use of world knowledge by prompting general-purpose Large Multimodal Models (LMMs) to augment task-specific visual representation with linguistic facial expression descriptions. A knowledge retriever is particularly designed to evaluate and select the most proper deliverable of LMMs.

## 2 RELATED WORK

### 2.1 Video-based Stress Detection by Applying Deep Learning Methods

Deep learning methods have achieved impressive results in various video-related tasks. There have been early efforts in video-based stress detection tasks. Jeon et al. [20] proposed a method for stress detection combining spatial and temporal attention. Its core idea is that the two attention mechanisms enable the model to focus on frames highly correlated with stress and the regions on a single frame that are highly correlated with stress. Gao et al. [13], Zhang et al. [48] correlated emotion with stress detection, and concluded that stress is sensitive to anger, fear and sadness. In this light, they proposed to classify a video as stressful when the percentage of expressions related to negative emotions in the video surpass a threshold. Furthermore, Zhang et al. [47] combined the dynamics of action with emotion for stress detection, introducing a two-leveled stress detection network named TSDNET. They utilized facial emotion recognition to obtain the most expressive and most expressionless facial images in the video to generate emotions representations, and obtained action representations by computing optical flow between the first and the last video frame. Despite the success in integrating action representations with emotion representations for stress detection, human action and stress-related emotion tend to exhibit different pace of shifts [24, 31]. In this light, we model action and emotion with

lens of different scales. Specifically, we propose Slow-Emotion-Fast-Action with a temporally varying dual-stream structure, in which dynamics of actions and emotions are encoded with a different pace. Meanwhile, in prior psychological studies, facial expressions can be decomposed into a group of units, in which certain co-occurrence of units can reveal states of inner feeling [5]. We thus delve into the facial expressions alongside action and emotion representations.

### 2.2 Large Multimodal Models

Large Multimodal Models (LMMs) represent the cutting-edge advances in artificial intelligence, designed to comprehend a diverse range of modalities, including text, images, audio, and video. LMMs are extensively pre-trained on large, diverse datasets that contain a variety of genres. Such pre-training process helps the model encode a wealth of concepts and cross-modal relationships, which lays the foundation for the downstream tasks. LMMs represented by GPT-4V have demonstrated stunning performance in image captioning and visual question answering [38]. Very recently, Wu et al. [43], Yang et al. [46] have successfully employed GPT-4V to tackle visual understanding, language comprehension, and visual puzzle solving tasks. In this work, we delve into the capability of LMMs to recognize facial expressions incorporating psychological FACS knowledge, which enhances the representation of stress.

## 3 METHODS

### 3.1 Overall Framework

Given a video $V$ of a subject, we aim to detect whether the subject is stressed or not. To capture the body actions of the subject, we follow Feichtenhofer et al. [12] to set a sampling rate $1/\tau$ that uniformly samples one frame out of $\tau$ frames from video $V$, obtaining a frame sequence named "Action Stream" denoted as $S_a = \{A_1, A_2, \cdots, A_n\}$, in which $n$ is the number of frames sampled. As the shift of human emotions exhibits a slower pace than actions, we set a slower sampling rate $1/m\tau$ to capture the details of emotions, in which $m$ is the sampling rate ratio. Specifically, for $S_a$ of $n$ frames, we analyze the subject's emotion based on a segment of $m$ ($m << n$) consecutive frames at each time, thus obtaining a set of $\lfloor n/m \rfloor$ emotion features of this subject. For each segment of $m$ frames, we design an emotion extraction module to obtain the representative emotion of the subject and the most emotional face image. After gathering emotion features from $\lfloor n/m \rfloor$ segments, we acquire a stream of subjects' emotions (referred to as "Emotion Stream") and a stream of the most emotional facial images (referred to as "Face Stream"), denoted as $S_e = \{E_1, E_2, \cdots, E_{\lfloor n/m \rfloor}\}$ and $S_f = \{F_1, F_2, \cdots, F_{\lfloor n/m \rfloor}\}$ respectively.

To construct visual representations for each video, we introduce the Slowfast mechanism to model the "Face Stream" and "Action Stream" jointly. Meanwhile, to enrich facial features while dealing with subtle differences in facial expressions, we introduce textualized facial description features by employing a large multimodal model (i.e., GPT-4V [1]) and pre-trained facial analysis model (i.e., SCN [39]) based on the "Face Stream". A sequence-level encoder then generates linguistic representation of the video after acquiring segment-level linguistic facial description features. Finally, we concatenate the visual and linguistic representation and employ a

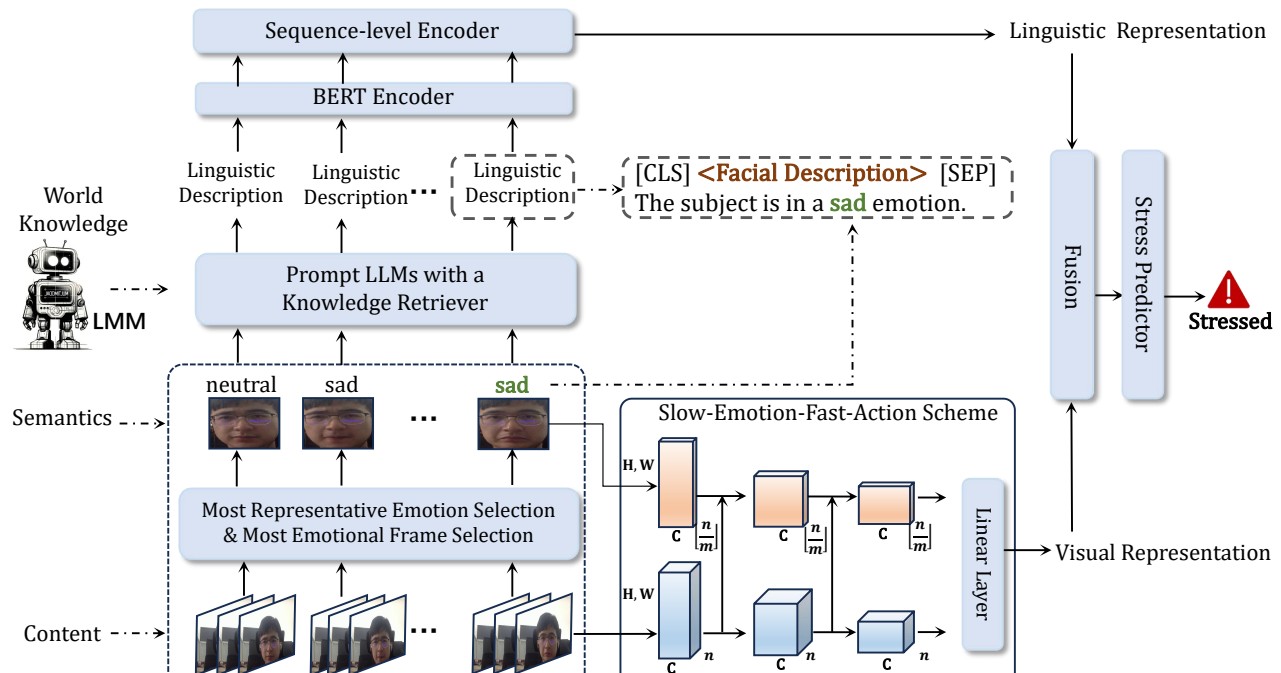

**Figure 1: Overall Framework of our three-level content-semantic-world knowledge framework. In the lower right part, $n$ and $\frac{n}{m}$ represent the length of sampled frames in fast pathway and slow pathway. $H$ and $W$ represent the height and width of each frame, and $C$ represents the number of channels.**

fully-connected stress predictor to detect the stress of the subject. The overall framework is illustrated in Figure 1.

## 3.2 Emotion Extraction for Video Segment

For each segment of $m$ consecutive frames from video $V$, denoted as $S_a^{i,i+m} = \{A_i, A_{i+1}, \cdots, A_{i+m}\}$, we employ a selection process that extracts the most representative emotion $E$, and the most emotional face image $F$, details shown below.

**Image-level Emotion Detection.** We adopt a facial recognition algorithm MTCNN [44] to localize the facial regions of each frame to obtain the collection of face images, denoted as

$$Face_{i,i+m} = \{face_i, face_{i+1}, \cdots, face_{i+m}\}.$$

Then we employ a Self-Cure Network (SCN) [39] to recognize the emotion of each face image with the form of seven categorical emotions (i.e., "angry", "disgust", "fear", "happy", "sad","surprise", "neutral"), and obtain the sequence of emotions, denoted as

$$Emotion_{i,i+m} = \{emo_i(p_i), emo_{i+1}(p_{i+1}), \cdots, emo_{i+m}(p_{i+m})\},$$

where $p_k(k = i, \cdots, i + m) \in [0, 1]$ is the confidence of the recognized emotion $emo_i$.

**Most Representative Emotion Selection.** In order to minimize the noise introduced by the emotion recognition model's misjudgment of individual frames, we select the emotion that appears most frequently in the $Emotion_{i,i+m}$ as the most representative emotion in this phase. It can be denoted as

$$E = MODE(Emotion_{i,i+m}),$$

where $MODE$ is the mode of the set.

**Most Emotional Face Image.** After obtaining the most representative expression $E$, we query in the $Emotion$ for the elements with $emo$ value $E$, next we select the element with the highest $p$ value, and define the face image corresponding to this element in $Face$ as the most emotional face image.

Here is an example with a segment of 6 face images, the sequence of emotion and the sequence of confidence obtained by SCN are $Emotion = \{$ "happy"(0.8982),"happy" (0.8803), "neutral"(0.7655), "surprise"(0.8763), "happy"(0.9866), "happy"(0.9951)$\}$. First, we obtain the most representative emotion $E =$"happy" by calculating the mode of the $Emotion$, and then we select the element from all "happy" elements with the highest confidence, so the last face image recognized as "happy" with confidence 0.9951 is selected as the most emotional face image.

## 3.3 Visual Encoding with Slow-Emotion-Fast-Action

As the shift of actions and emotions exhibits different paces, we propose Slow-Emotion-Fast-Action framework inspired by [12], which learns the visual representation of a video through a temporally varying dual-stream structure. Specifically, the slow pathway has larger channel sizes and lower frame rate to capture subtle details of emotions, while the fast pathway has high temporal resolution to capture fast body motions. Therefore, we adopt $S_f$ as the input of the slow pathway, and the larger size of channels can obtain more semantic information for each face image. On the other hand, we take $S_a$ as

> **Facial Description**
>
> **Prompt:**
> Describe the facial movements including **Eyes**, **Nose**, **Lid**, **Lips**, **Lip Corner**, **Neck**, **Jaw**, **Cheek**, **Chin**, **Mouth**, **Dimple**, **Tongue** and **Eyebrow** of the character in the photo.
>
> 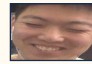

**Figure 2: Our prompt that guides the LMM to generate facial expression descriptions underlining significant face regions based on FACS knowledge.**

the input of fast pathway to capture the change of movement with higher temporal resolution. The visual representations of video $V$ are computed as follows:

$$R_V = Slowfast(S_f, S_a).$$

The details of the slow pathway and the fast pathway are shown in Table 2.

## 3.4 Linguistic Encoding via Knowledge from LMMs

We further exploit knowledge from large multimodal models (LMMs) to enrich semantics conveyed in "Emotion Stream" $S_e = \{E_1, E_2, \cdots, E_{n/m}\}$ and "Face Stream" $S_f = \{F_1, F_2, \cdots, F_{n/m}\}$. For the $i$-th segment, the most representative $E_i$ is combined with the most emotional face image $F_i$ and fed into the segment-level encoder to acquire linguistic representation. Given $\lfloor n/m \rfloor$ segment-level linguistic representations, sequence-level linguistic representation of the whole video can be generated via the sequence-level encoder.

**Facial Description Generation and Selection**. Large multimodal models (LMMs) have demonstrated the capacity to recognize fine-grained visual features [26, 50]. For face image $F_i$, we employ a LMM (i.e., GPT-4V) to recognize facial expressions with an instruction based on Facial Action Coding System (FACS). FACS is a widely applied manual extraction method for facial feature extraction, which divides facial expressions into 46 primary units, focusing on 9 regions (i.e., eyes, nose, lid, lips, lip corner, neck, jaw, cheek and eyebrow). We thus instruct the LMM to focus on these 9 regions and describe corresponding movements, the instruction and result shown in Figure 2.

Unfortunately, the outputs of LMM can be diverse, including unfaithful content that does not meet the demand due to hallucination [28, 49]. Therefore, for each face image $F_i$ we sample $k$ times to obtain $k$ different descriptions from the LMMs, denoted as $\{d_1, \cdots, d_k\}$. We design a FACS knowledge retriever $R$ that selects the best output among these descriptions, which is a vision-language model consisting a linguisticencoder $\mathcal{T}$ and image encoder $\mathcal{I}$. For each LMM description $d_j$ it generates a score: $R(d_j, F_i) = sim(\mathcal{T}(d_j), \mathcal{I}(F_i))$, where $sim(\cdot, \cdot)$ denotes cosine similarity, $\mathcal{T}(d_j)$ and $\mathcal{I}(F_i)$ denotes linguistic representation of $d_j$ and visual representation of $F_i$ encoded by $\mathcal{T}$ and $\mathcal{I}$. As a result, the final facial description for the $i$-th segment is selected as:

$$D_i = \operatorname{argmax}_{d_j \in \{d_1, \cdots, d_k\}} R(d_j, F_i).$$

To help the retriever $R$ effectively distinguish helpful LMM answer from unfaithful ones, we train $R$ before stress detection learning. We build a face image set from the training samples of stress detection, for each face image $F_i$, we employ expert to annotate facial action units based on FACS knowledge, and transform them into natural language descriptions as ground truth $\hat{d}$. We compute linguisticsimilarity between $\hat{d}$ and $d_1, \cdots, d_k$, which are $k$ answers from LMMs to describe facial expressions of $F_i$. We label the answer with highest similarity with $\hat{d}$ as positive and $k'$ answers with lowest similarity as negative, denoted as $d^+$ and $d_1^-, \cdots, d_{k'}^-$ respectively. We can now define the typical contrastive learning objective and minimize for each face image the negative log likelihood of the positive example:

$$\mathcal{L}(F_i, d^+, d_1^-, \ldots, d_{k'}^-)$$
$$= -\log \frac{e^{sim(\mathcal{I}(F_i), \mathcal{T}(d^+))}}{e^{sim(\mathcal{I}(F_i), \mathcal{T}(d^+))} + \sum_{j=1}^{k'} e^{sim(\mathcal{I}(F_i), \mathcal{T}(d_j^-))}}. \quad (1)$$

**Segment-level Encoding**. After retrieving the best facial expression description with $R$, we construct a manual template to integrate the emotion $E_i$ with facial expressions generated by LMMs in a natural language form, which can be fed into the BERT encoder. The template is shown in Table 1.

**Table 1: Manual Template to integrate emotion $E_i$ with facial expressions $D_i$. [CLS] and [SEP] are special tokens of BERT for representing sentence beginnings and segmenting sentences, respectively.**

| Manual Template |
| --- |
| [CLS] $\langle D_i \rangle$ [SEP] This subject is in a $\langle E_i \rangle$ emotion. |

We denote the above text integrating $D_i$ and $E_i$ as $T_i$. We then feed $T_i$ into a trainable BERT [7] encoder to obtain segment-level representations $x_i$.

**Video-level linguistic representation**. Given the linguistic representation of segment-level facial description $x_i$, we feed the representation of all phases $X = \{x_1, x_2, \cdots, x_{\lfloor n/m \rfloor}\}$ into a Long Short-Term Memory network (LSTM) to acquire the hidden states in the sequence:

$$h_t = LSTM(x_t, h_{t-1}),$$

where $h_t$ and $h_{t-1}$ denote the hidden state vector of LSTM at time $t$ and $t-1$. In addition, we aggregate the hidden states from all phases via a phase attention layer to obtain the attention vector $\alpha$ that assign different weights to each phase.

$$\alpha = softmax(\tilde{h} \times w_a + b_a),$$

where $\tilde{h} = \{h_1, h_2, \cdots, h_{n/m}\}$ is the aggregation of the hidden states from all phases, $w_a$ and $b_a$ are trainable parameters of the phase attention layer. Finally, we obtain the video-level linguistic representation $R_T$ of the video:

$$R_T = (\alpha^T \times \tilde{h}) \times w_t + b_t,$$

where $w_t$ and $b_t$ are trainable parameters.

**Table 2: Architecture of the Slowfast with ResNet-18 backbone. Strides are denoted as $\{temporal\ stride, spatial\ stride^2\}$, the dimensions of kernels are denoted by $\{T \times S^2, C\}$ for temporal, spatial and channel sizes. The input of slow pathway is Face Stream $S_f$, the input of fast pathway is Action Stream $S_a$.**

| Stage | *Slow* pathway | *Fast* pathway | Output sizes $T \times S^2$ |
|---|---|---|---|
| raw input | Face Stream $S_f$ | Action Stream $S_a$ | $Slow: 8 \times 224^2$ 
 $Fast: 64 \times 224^2$ |
| $conv_1$ | $1 \times 7^2$, 64 
 stride 1, $2^2$ | $5 \times 7^2$, 8 
 stride 1, $2^2$ | $Slow: 8 \times 112^2$ 
 $Fast: 64 \times 112^2$ |
| $pool_1$ | $1 \times 3^2$ max 
 stride 1, $2^2$ | $1 \times 3^2$ max 
 stride 1, $2^2$ | $Slow: 8 \times 56^2$ 
 $Fast: 64 \times 56^2$ |
| $res_2$ | $\begin{bmatrix} 1 \times 3^2, 64 \\ 1 \times 3^2, 64 \end{bmatrix} \times 2$ | $\begin{bmatrix} 3 \times 3^2, 64 \\ 1 \times 3^2, 64 \end{bmatrix} \times 2$ | $Slow: 8 \times 56^2$ 
 $Fast: 64 \times 56^2$ |
| $res_3$ | $\begin{bmatrix} 1 \times 3^2, 128 \\ 1 \times 3^2, 128 \end{bmatrix} \times 2$ | $\begin{bmatrix} 3 \times 3^2, 64 \\ 1 \times 3^2, 64 \end{bmatrix} \times 2$ | $Slow: 8 \times 28^2$ 
 $Fast: 64 \times 28^2$ |
| $res_4$ | $\begin{bmatrix} 3 \times 3^2, 256 \\ 1 \times 3^2, 256 \end{bmatrix} \times 2$ | $\begin{bmatrix} 3 \times 3^2, 256 \\ 1 \times 3^2, 256 \end{bmatrix} \times 2$ | $Slow: 8 \times 14^2$ 
 $Fast: 64 \times 14^2$ |
| $res_5$ | $\begin{bmatrix} 3 \times 3^2, 512 \\ 1 \times 3^2, 512 \end{bmatrix} \times 2$ | $\begin{bmatrix} 3 \times 3^2, 512 \\ 1 \times 3^2, 512 \end{bmatrix} \times 2$ | $Slow: 8 \times 7^2$ 
 $Fast: 64 \times 7^2$ |

## 3.5 Stress Detection

After we acquire the visual and linguistic representations of the video, we fuse the two representations and classify the stress of the subjects in the video. We merge visual representation $R_V$ and global linguistic representation $R_T$, and then apply a fully connected stress predictor $p$ to compute whether the subject is under stress:

$$\hat{y} = \text{Sigmoid}(p(R_V \oplus R_T)).$$

Given stress label $y$, we optimize the detection framework with the following cross entropy loss:

$$\mathcal{L}(\hat{y}, y) = -[y \log \hat{y} + (1-y) \log(1-\hat{y})]. \tag{2}$$

## 4 EXPERIMENT

### 4.1 Datasets.

*4.1.1 UVSD Dataset.* The UVSD dataset [47] comprises 490 videos, each approximately two minutes in duration, featuring 112 college student subjects (58 males and 64 famales aged 18-26). Subjects were asked to watch videos while being recorded by a video camera. Videos were labeled as "unstressed" when subjects were asked to watch videos of scenery, food production, and variety show episodes, and "stressed" when subjects were asked to watch knowledge-intensive videos and take a question and answer test after they finished watching. The dataset contains a total number of 2,092 video samples, including 920 "stressed" samples, and 1,172 "unstressed" samples.

*4.1.2 Reality Show about Lies (RSL) Dataset.* Lying is a complex psychological behavior intricately tied to cognitive processes and mental activities [6]. For most people without special training, lying induces psychological stress which can be reflected in facial expressions, sounds and body movements [34]. Drawing inspiration from this, we curated a video stress dataset based on a reality TV program named "Odd Man Out". In an episode titled "6 Introverts vs 1 Secret Extrovert," for instance, an extrovert impersonates an

**Table 3: Dataset Statistics of UVSD and RSL.**

| | UVSD | | RSL | |
|---|---|---|---|---|
| | #Clips | Avg. Duration | #Clips | Avg. Duration |
| Stressed | 920 | 15.00s | 132 | 5.89s |
| Unstressed | 1172 | 15.00s | 313 | 6.53s |
| Total | 2092 | 15.00s | 445 | 6.34s |

introvert by disguising his or her identity as an extrovert as much as possible in conversations and Q&A sessions with six introverts. Throughout each episode, one or more individuals assume deceptive roles, trying to seamlessly blend into specified categories relevant to the episode's theme. The show incorporates multiple voting sessions, with the participant garnering the most votes being eliminated. Surviving until the end, the liar(s) stand to claim the prize money, motivating heightened engagement throughout the program. Encountering constant scrutiny, the liar(s) undergo persistent stress from beginning to the end. All video frames featuring the liar(s) were labeled as "stressed" except for the end-of-show statement. Conversely, the end-of-show statement by the liar(s) and all footage of other participants were labeled as "unstressed".

To construct the dataset, we selected 30 truth-tellers and 30 liars from a total of 195 participants over 28 sessions, maintaining a one-to-one male-to-female ratio. All video frames were refined to depict only one participant per frame, with clear visibility of their face and upper body. This meticulous process yielded a dataset comprising 706 video clips, further detailed in Table 3 as the Reality Show about Lies (RSL) Dataset.

### 4.2 Experiment Settings

For all experiments, we perform 10-fold cross validation. The original format of image frames from UVSD and RSL dataset is $640 \times 480$ ($height \times width$). We resize each original frame into $224 \times 224$, and feed them into the model. The format of each face image obtained by the MTCNN is also $224 \times 224$. We leverage Resnet-18 as the backbone network structure in visual representations with Slow-Emotion-Fast Action. we set the sampling rate $1/\tau$ to 1/8 following Feichtenhofer et al. [12], and the sampling ratio $m$ to 8. More details of the slow pathway and fast pathway are shown in Table 2. We adopt *gpt-4-vision-preview* from the GPT-4 family for facial description, and we also limit the token number of model output within 150. We employ *CLIP-B/16* (CLIP-B) [33] as the knowledge retriever $R$. Specifically, it consists of a *ViT-B/16* Transformer as the image encoder and a masked self-attention Transformer [36] as the text encoder. To collect training samples for retriever $R$, we annotate AU labels with 300 face images, which are randomly selected from 150 stressed video samples and 150 unstressed video samples of UVSD dataset. The maximum sentence length of the information integration is set to 200, including the [CLS] token at the beginning of the sentence and the [SEP] token prior to the facial description. To optimize the knowledge retriever with loss term Eq. 1, we set the learning rate to 3e-5 and the batch size to 32. We set the number of negative samples $k'$ to 3. To optimize the detection model with loss term Eq. 2, we set the batch size to 16 and learning rate to 3e-3. Adam [21] is adopted as the optimizer.

**Table 4: Stress detection performance of our method and all competitive baselines over UVSD and RSL dataset.**

| Categories | Method | UVSD | | | | RSL | | | |
|---|---|---|---|---|---|---|---|---|---|
| | | Acc. | Prec. | Rec. | F1. | Acc. | Prec. | Rec. | F1. |
| Action Units based Stress Detection | Dependent Model [37] | 70.46% | 70.28% | 71.39% | 71.08% | 64.04% | 58.90% | 59.76% | 59.33% |
| | FDASSNN [14] | 74.11% | 73.71% | 74.00% | 74.06% | 67.42% | 62.26% | 63.26% | 62.75% |
| Frame-wise Emotion Detection and Aggregation | Gao et al. [13] | 78.38% | 65.00% | 63.83% | 64.40% | 63.30% | 52.81% | 62.42% | 52.61% |
| | Zhang et al. [48] | 81.58% | 67.38% | 77.30% | 72.00% | 65.49% | 56.77% | 56.21% | 56.49% |
| Video-wise Stress Detection | Jeon et al. [20] | 82.71% | 69.61% | 77.30% | 73.26% | 79.53% | 74.54% | 64.72% | 66.78% |
| | TSDNet [47] | 85.42% | 85.28% | 85.32% | 85.53% | 81.76% | 80.37% | 72.77% | 74.99% |
| | GPT-4V [1] | 75.62% | 75.21% | 75.72% | 75.31% | 69.21% | 64.37% | 65.63% | 64.99% |
| Ours | | **91.25%** | **92.18%** | **90.24%** | **90.89%** | **86.50%** | **84.81%** | **78.40%** | **80.79%** |

## 4.3 Baseline Methods

We categorize the currently competitive video-based stress detection methods into three groups, and then compare our method with representative methods from each category.

**(1) Action units-based stress detection**

**Dependent Model**. Viegas et al. [37] propose to extract 17 different action units from the subject's face image, and then employ different simple classifiers (i.e. Random Forest, LDA, Gaussian Naive Bayes and Decision Tree) to recognize stress based on the action units. We implement this baseline using the best performing Gaussian Naive Bayes classifier.

**FDASSNN**. Gavrilescu and Vizireanu [14] employ an Active Appearance Model (AAM) [9] to detect intensities of different action units, and a multi-layer perceptron to transform action unit intensities into stress detection result.

**(2) Frame-wise Emotion Detection and Aggregation**

**Gao et al.** Gao et al. [13] extract 49 feature points of each face image from a video and apply SVM to classify each frame as positive or negative emotions. When the percentage of frames with negative emotions in a video exceeded a threshold, the video can be classified as stressed.

**Zhang et al**. Zhang et al. [48] leverage a Convolutional Neural Network (CNN) to detect emotion in each video frame. If two-thirds of the frames show emotions of anger, sadness, or fear, the video can be detected as having stress.

**(3) Video-wise Stress Detection**.

**Jeon et al**. Jeon et al. [20] incorporate the features of each video frame encoded by ResNet-18 and the features of facial landmarks encoded by a Facial Landmark Feature Network to form frame-level representations. A temporal attention module further incorporates frame-level representations of each video into stress detection.

**TSDNet**. Zhang et al. [47] first obtain face- and action-level representations separately, and then fuse the results through a stream-weighted integrator with local and global attention for video stress detection.

**GPT-4V**. GPT-4-vision [1] (GPT-4V) is a state-of-the-art large multimodal model. GPT-4V demonstrates impressive vision understanding ability, yet it does not possess the ability to accept video

**Table 5: Performance of our method using different modalities.**

| Dataset | Modality | Acc. | Prec. | Rec. | F1. |
|---|---|---|---|---|---|
| UVSD | only vision | 87.81% | 87.79% | 87.23% | 87.47% |
| | only linguistic | 75.63% | 75.43% | 75.42% | 75.44% |
| | Ours | **91.25%** | **92.18%** | **90.24%** | **90.89%** |
| RSL | only vision | 85.52% | 82.21% | 73.81% | 76.68% |
| | only linguistic | 68.75% | 68.89% | 69.07% | 68.71% |
| | Ours | **86.50%** | **84.81%** | **78.40%** | **80.79%** |

data directly. To implement video-based stress detection with GPT-4V, we take 10 frames evenly from each sample and allow the GPT-4V to determine the stress labels of the videos.

## 4.4 Performance

As shown in Table 4, our framework outperforms others on the UVSD dataset, achieving the highest accuracy (91.25%) and F1-score (90.89%). This represents a substantial improvement of over 4.83% and 5.36% compared to TSDNet [47], the best video-wise stress detection baseline. Additionally, compared to the leading frame-wise emotion detection and aggregation method by Zhang et al. [48], our method demonstrates improvements of over 9.77% and 18.8%. When contrasted with machine learning-based methods like Dependent Model [37] and FDASSNN [14] that rely on action units, the enhancements our method brings are even more pronounced. In the RSL dataset, our framework also achieves the best performance, with 86.50% accuracy and 80.79% precision, outperforming the best baselines such as TSDNet [47] by 4.74% and 5.80%, respectively. These all results confirm the effectiveness of our framework in incorporating the Slow-Emotion-Fast-Action scheme, FACS, and LMM for stress detection.

## 4.5 Study of Our Method

*4.5.1 Effectiveness of Visual Encoding and Linguistic Encoding.* To evaluate the impact of different modalities, we conducted a modality ablation study with ablation variant **"only vision"** that only uses visual representations for stress detection, and **"only linguistic"** that only uses linguistic representations for stress detection. The

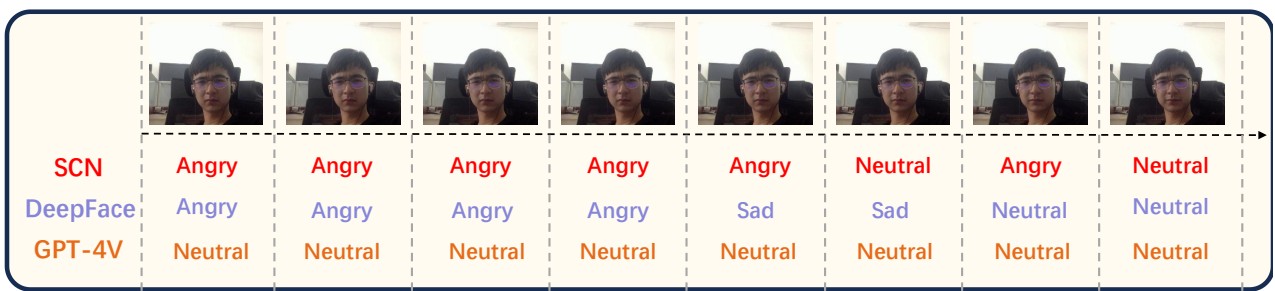

**Figure 3: Case study on noise introduced by emotion classifiers.**

**Table 6: Comparison of the effectiveness of facial description representation using the large model and FACS.**

| Dataset | Method | Acc. | Prec. | Rec. | F1. |
|---------|--------|------|-------|------|-----|
| UVSD | AUs | 88.75% | 88.71% | 88.63% | 88.64% |
| | LMMs (Ours) | **91.25%** | **92.18%** | **90.24%** | **90.89%** |
| RSL | AUs | 86.25% | 82.49% | 75.36% | 77.24% |
| | LMMs (Ours) | **86.50%** | **84.81%** | **78.40%** | **80.79%** |

results are presented in Table 5. We observed a performance decline of 3.42% and 4.11%, when excluding the text modality. Similarly, excluding the vision modality led to a decline of 15.4% and 12.0%. These results highlight the greater contribution of the visual modality to our model compared to the linguistic modality.

*4.5.2 The effectiveness of leveraging LMMs to generate facial description.* We validate the effectiveness of our facial description generation by prompting LMMs, compared with directly using Action Units (AUs) as facial description. We design ablation variant **"AUs"** which replaces our linguistic representation with AU-based representation by detecting AUs on each frame and transforming the AU signals into a video-level representation with a LSTM encoder [29]. As shown in Table 6, using the rich linguistic information obtained from the LMMs for facial description provided stronger representation information than action units with only discrete AU signals in the stress detection task, this mechanism enhances performance by 2.25% and 3.55% on the respective datasets.

*4.5.3 Effectiveness of Slow-Emotion-Fast-Action.* To evaluate the rationality of the Slow-Emotion-Fast-Action scheme, we conducted experiments focusing on the simultaneous consideration of emotion and action at the same sampling frequency. We design ablation variants with pure slow or fast sampling rates. Specifically, **1. "SE-SA"** represents temporal invariant dual-stream structure with slowpath, and **2. "FE-FA"** represents temporal invariant dual-stream structure with fastpath. The results are presented in Table 7. When both emotion and action use a slow sampling frequency, the performance levels are 80.53% and 68.32%, respectively. Conversely, using a fast sampling frequency for both yields performance levels of 86.56% and 76.68%. However, employing the Slow-Emotion-Fast-Action scheme leads to the highest performance of 90.89% and 80.79%, underscoring the rationale and effectiveness of this approach.

**Table 7: Performance of our method and ablation variants with different visual encoding.**

| Dataset | Method | Acc. | Prec. | Rec. | F1. |
|---------|--------|------|-------|------|-----|
| UVSD | SE-SA | 80.63% | 80.67% | 81.32% | 80.53% |
| | FE-FA | 87.19% | 88.29% | 85.86% | 86.56% |
| | SE-FA (Ours) | **91.25%** | **92.18%** | **90.24%** | **90.89%** |
| RSL | SE-SA | 81.50% | 81.26% | 65.66% | 68.32% |
| | FE-FA | 85.50% | 82.20% | 73.81% | 76.68% |
| | SE-FA (Ours) | **86.50%** | **84.81%** | **78.40%** | **80.79%** |

**Table 8: Performance of our method and ablation variants with different linguistic encoding.**

| Dataset | Method | Acc. | Prec. | Rec. | F1. |
|---------|--------|------|-------|------|-----|
| UVSD | w/o FACS | 89.06% | 89.19% | 88.42% | 88.73% |
| | gen. only | 89.68% | 89.76% | 89.08% | 89.36% |
| | Ours | **91.25%** | **92.18%** | **90.24%** | **90.89%** |
| RSL | w/o FACS | 84.68% | 83.26% | 74.45% | 77.48% |
| | gen. only | 85.44% | 83.79% | 76.48% | 79.05% |
| | Ours | **86.50%** | **84.81%** | **78.40%** | **80.79%** |

*4.5.4 Effectiveness of FACS knowledge.* To evaluate the impact of FACS knowledge in our method, which guides the LMM to generate facial descriptions of crucial facial regions, and develops a knowledge retriever selecting most proper deliverable of LMMs, we design the following ablation variants: **1. "w/o FACS"** guides the LMM to generate facial descriptions with a simple prompt: "Describe the facial movements of the character in the photo", and removes the knowledge retriever, using a random generation as final description. **2. "gen. only"** only removes the knowledge retriever. As depicted in Table 8, when FACS knowledge is not used ("w/o FACS"), the F1-score 88.73% and 77.48%. Conversely, when only using FACS knowledge for prompting without the knowledge retriever ("gen. only"), performance increased to 89.36% and 79.05%. Furthermore, incorporating the knowledge retriever results in a performance boost to 90.89% and 80.79%. These results demonstrate the significance of our scheme that incorporates FACS knowledge to better generate linguistic facial expression descriptions with LMMs.

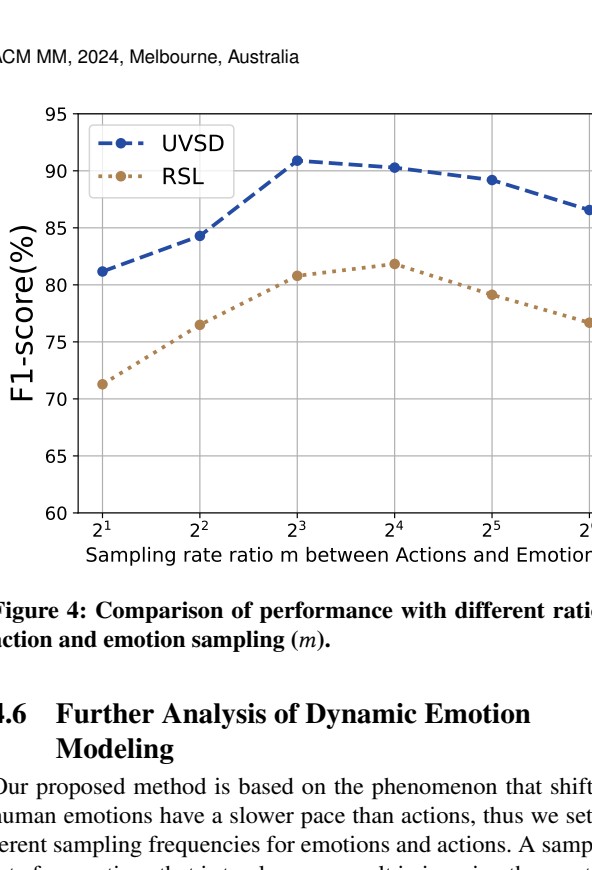

**Figure 4: Comparison of performance with different ratio of action and emotion sampling ($m$).**

## 4.6 Further Analysis of Dynamic Emotion Modeling

Our proposed method is based on the phenomenon that shifts of human emotions have a slower pace than actions, thus we set different sampling frequencies for emotions and actions. A sampling rate for emotions that is too low can result in ignoring the emotions that actually occur, and a sampling frequency that is too high can suffer from noise interference due to problems with the accuracy of emotion classifiers. As is illustrated in Figure 3, we enumerate the outcome of SCN, one of the most effective emotion recognition models, Deepface [35], the most commonly used facial emotion recognition tool, and GPT-4V for emotion recognition on consecutive video frames. In addition to the significant lack of ability of GPT-4V in emotion recognition [27], the other two dedicated models are unable to maintain consistent results in frame-by-frame emotion detection with variations and their subtleties.

We empirically verify the impact of different sampling ratios on the experimental results. As shown in Figure 4, the best result of 90.89% is obtained when the ratio reach to $2^3$ on the UNVD dataset and 81.83% at $2^4$ on the RSL dataset. Before obtaining the best result, the model's performance on both datasets increase considerably with the rise of the sampling ratio, with a comparison of the performances at a sampling ratio of $2^1$ increasing by 9.72% and 10.55%, respectively. Thereafter the model's performance on both datasets start to decrease with the growth of the sampling ratio, decreasing by 4.33% and 5.15% when the sampling ratio reaches $2^6$, respectively.

## 4.7 Case study on Facial Description Generation Methods

We provide a case study to analyze our prompting LMMs to generate facial descriptions and selecting the best generation based on FACS knowledge. As shown in Table 9, the LMMs output with the FACS-based prompt and retriever (see "with FACS Knowl.") fully encompasses the knowledge that the Action Units can provide, and also provides more detailed description of the face region by region.

**Table 9: Case study on facial description generation methods, the text marked in teal is the part that agrees with the content of the action unit representation.**

| Face Image | Action Units |
|---|---|
| | ✓AU6 Cheek Raiser
✓AU7 Lid Tightener
✓AU20 Lip Stretcher
✓AU27 Mouth Stretcher
✓AU42 Slit |

| Facial description using Large Multimodal Models (LMMs) | |
|---|---|
| w/o FACS Knowl. | with FACS Knowl. |
| The character in the photo is showing a broad smile with their eyes closed. The smile leads to the cheeks being *raised*[6] and a bunching of the skin around the eyes, which is often associated with a genuine, joyful smile. This facial expression typically indicates happiness or amusement. | **Eyes:** The eyes are *narrowed and slightly creased*[42] at the corners, which is typical when a person is smiling or laughing. **Check:** The cheeks are *raised*[6], pushing up towards the eyes, which contributes to the squinting appearance of the eyes. **Lid:** The eyelids are partially *closed*[7], as they often are when a person smiles broadly. **Lips:** The lips are *parted and stretched*[20] horizontally, revealing the upper teeth in a smiling gesture. **Mouth:** *Open*[27] in a wide smile, showing teeth, which suggests happiness or laughter... |

## 5 CONCLUSION

This paper shows an effective way to integrate task-specific video-based stress detection model with general purpose Large Multimodal Models to achieve strong performance. Specifically, we introduce a three-level content-semantic-world knowledge framework for video-based stress detection. A Slow-Emotion-Fast-Action scheme is designed to encode temporal changes of body actions as well as subtle details of emotions. It explores the use of world knowledge by prompting general-purpose large multimodal models (LMMs) to generate linguistic facial expression descriptions. Furthermore, we capitalize on Facial Action Coding System (FACS) knowledge to design a prompt to instruct the LMMs and develop a knowledge retriever to select the most proper deliverable of LMMs. The experiments on two datasets show that our framework achieves state-of-art performance.

## 6 LIMITATIONS

We identify one major limitation of this work is its input modality. Specifically, our method is limited to detecting stress with video inputs and ignores inputs in other modalities such as audio and electroencephalogram (EEG). Such modalities provide valuable information that can be used to enhance stress detection. Fortunately, through multi-modal pre-training models [18, 45], we can obtain robust representations with more diverse modalities.

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
