# OpenReview forum: "Integrating Content-Semantics-World Knowledge to Detect Stress from Videos"
_acmmm.org/ACMMM/2024/Conference — MM2024 Poster_

### Official Review · Reviewer_1vpk · 2024-05-23

**Rating:** 4
**Confidence:** 4

**Summary:**

This paper proposes a video-based stress detection method. The authors design a Slow-Emotion-Fast-Action scheme to encode fast changes of body actions, as well as subtle details of emotions into visual representation. Then, they augment the derived visual representation with linguistic facial expression descriptions by prompting general-purpose Large Multimodal Models (LMMs). Considering the potentially unfaithful descriptions, a knowledge retriever is designed to evaluate and select the most proper deliverable of LMMs. Experiments show state-of-the-art performance on two benchmark datasets.

**Strengths:**

The paper is well-organized and easy to follow.

The idea is natural and technically sound. I like the interpretability of the intermediate results of the model.

Extensive experiments are conducted, verifying most statements in the introduction and method sections. The improvements look solid. I like the insightful discussions in the experiments.

**Limitations:**

I believe this paper is above the borderline, although the following concerns prevent me from considering this paper for direct acceptance.

[INTRODUCTION] Some motivations are not clear.

The authors claim they build a "three-level (content-semantic-world knowledge) framework". I am confused about what semantics and world knowledge correspond to after reading the paper. Do they refer to facial expression descriptions and the knowledge retriever? Could the authors reformulate lines 108-120 to clarify this?

I notice that the authors adopt several pre-trained models, e.g., the facial recognition algorithm MTCNN, Self-Cure Network (SCN) [39] to recognize the emotion of each face image in the form of seven categorical emotions, and GPT-4V to recognize facial expressions with an instruction based on the Facial Action Coding System (FACS). I wonder if the proposed method majorly benefits from different pre-trained models rather than the designed pipeline.

In this case, fine-tuning a pre-trained LMM on the datasets could achieve equivalent performance without the need for the pipeline, taking the predictions of smaller models as inputs in the prompt.

[INTRODUCTION] Some details are missing.

The complex process and lack of details may hinder the reproducibility of the method. I am curious about the network structure of the proposed method, such as the hidden dimensions in the sequence-level encoder. I also wonder if some details are omitted in the Slow-Emotion-Fast-Action Scheme and stress predictor, e.g., the settings of activation functions. I couldn't find the specification of some hyperparameters, e.g., the number of training epochs.

The authors claim they "employ experts to annotate facial action units based on FACS knowledge, and transform them into natural language descriptions as ground truth". More detailed statistics and examples of the annotated results should be provided.

[Experiments] Some experiments are omitted.

I like the design of "Most Representative Emotion Selection & Most Emotional Frame Selection", but the authors omit the ablation study. I wonder about the performance gain brought by the designed selection compared to sampling at fixed intervals or random sampling.

I wonder if the proposed method can only be used for full video recordings, which should contain more than 2^6 frames according to the setting of m in Fig. 4. It might be a rather impractical experimental setup that hinders real-time stress detection. I wonder how the performance varies with different numbers of frames, n.

**Suitability:**

3

---

### Official Review · Reviewer_tgRj · 2024-05-24

**Rating:** 3
**Confidence:** 3

**Summary:**

This paper studies the task of stress detection based on videos and proposes a three-tier Content-Semantic World Knowledge Framework. The innovation of this framework lies in leveraging the LMMs to enhance the capability of video frames in expressing emotional and linguistic facial features for the task. And it utilizes a knowledge retriever to evaluate and select the most suitable LMMs output representation. Moreover, through the Slow-Emotion-Fast-Action Scheme, it acquires visual representations capable of capturing both the speed of actions and subtle emotional details. Experimental results demonstrate that compared to other methods, this framework achieves superior performance in video-based stress recognition.

**Strengths:**

1. The article is well-written, clearly presenting the proposed framework and experimental results. The description of the methods is detailed, making it easy to understand the article's structure.
2. The article effectively leverages large-scale language models for application in video pressure detection tasks, yielding favorable results.
3. The experimental results of this approach are robust compared to other methods and include comprehensive elimination experiments.

**Limitations:**

1. The method proposed in the article is based on the phenomenon where human emotional changes are slower than actions. Is there existing research that serves as a foundation for this setup? Some emotional changes can also occur rapidly.
2. The facial description selection method mentioned in the article involves additional data annotation and training before the experimental training. This could significantly increase the training cost of the model.
3. The baseline method for experimental comparison is quite outdated. It would be beneficial to incorporate recent research articles for comparison.
4. The workflow of this framework is rather complex. Considering the complexity of the framework, it is advisable to include some experimental analyses.

**Suitability:**

3

---

### Official Review · Reviewer_M8Tf · 2024-05-25

**Rating:** 2
**Confidence:** 3

**Summary:**

This paper tackles the problem of stress detection from videos. Their system utilizes video frame features and language descriptions extracted via the video frames itself. To generate the natural language descriptions, authors extract representative emotion label $E_i$, one of seven discrete emotion labels, and a corresponding representative face image $F-i$. For $m$ segments within a video, $F_i$ serves as an input to LMM (GPT-4V) with additional context about the Facial Action Coding System (FACS). To mitigate hallucinations, authors sample $k$ different descriptions from LMMs and use an additional retriever $R$ (CLIP-B) to select the descriptions. The final facial description proposed by $R$ serves as input to the BERT-encoder, which yields the final segment-level representations. To obtain video-level linguistic representation, all the segment representations are fed into an LSTM model, and the final aggregated hidden state is extracted. Video features are processed via the Slow-Fast network. The representations extracted for both modalities are fused over which a fully connected predictor/network is applied for stress label classification. The authors conduct experiments on UVSD and the newly proposed RSL dataset.

**Strengths:**

1. Covers important experiments to justify the modalities, motivation to use the Slow-Fast approach, and effectiveness of LMMs to generate factual descriptions.
2. Well written paper and easy to follow.

**Limitations:**

1. The proposed method is computationally complex and performs stress detection in a very restrictive environment. The proposed approach cannot scale to in-the-wild settings with multiple people, narration, subject not facing the camera, etc. The restrictive setting does injustice to the potential of all the models necessary to the pipeline (GPT-4, CLIP-Base, BERT-encoder, MTCNN, SCN, Slow-Fast, LSTM, FN-Classifier).
2. For the proposed RSL dataset, claiming all the frames with liars as "stressed" is unfair. "Odd Man Out" TV show had many scenarios where actors laugh, act cool, and show no signs of stress.
3. Need additional details on GPT-4V baseline. Were the 10 frames provided to GPT-4V sampled from the subset used by the author's model? If different, how relevant were the frames for identifying stress from the video?
4. The justification for using a slow-fast network is not sufficient. The network expects the inputs to have different sampling rates, and therefore, modifying those is unfair. An appropriate experiment and comparison would be to replace the slow-fast backbone with any other backbone, such as ResNet50, etc. It would also be interesting to see a pre-trained emotion recognition backbone (ResNet50 pre-trained on FER13, SFEW, etc, datasets). Such pre-trained models will provide richer emotional representations.
5. Line 572, CLIP is limited to 77 input tokens; it is unclear how authors set the input sentence length to 200, which includes the [CLS] and [SEP] tokens.

**Suitability:**

2

---

### Official Review · Reviewer_5Mm8 · 2024-05-27

**Rating:** 6
**Confidence:** 4

**Summary:**

This paper proposes a content-semantic world knowledge framework for video-based stress detection. The authors design a Slow-Emotion-Fast-Action scheme to encode fast temporal changes of body actions revealed from video frames, as well as subtle details of emotions per video segment, into visual representation. They further augment task-specific visual representation with linguistic facial expression descriptions and introduce a knowledge retriever to evaluate and select the most proper deliverable of LMMs.

Empirical results how the effectiveness of the proposed method.

**Strengths:**

I appreciate this paper's effort to develop an interpretable approach with world knowledge for emotion detection. The proposed approach  is reasonable and is a good example for further research to instill knowledge into model design, instead of only relying on black-box models for decision making. Therefore, I recommend "Accept" for this paper to advocate the endeavors on this direction.

**Limitations:**

Have the facility descriptions generated by LMMs been evaluated by human experts? It would be more convincing to if the outputs from LMMs can be validated by human experts.

**Suitability:**

3

---

### Meta-Review · Area_Chair_aC3w · 2024-06-30

**Recommendation:** Accept (Poster)
**Confidence:** 4

**Metareview:**

The paper received mixed review. The paper is well-written, the motivation is clear and the approach is reasonably novel, which the reviewers agreed on. However, as reviewers pointed out, there are some limitations, and the design needs some more justification and more experimentation in the future.